# A geometrical connection between sparse and low-rank matrices and its application to manifold learning

**Lawrence K. Saul**  *lsaul@flatironinstitute.org*
*Center for Computational Mathematics*
*Flatiron Institute*
*162 Fifth Avenue*
*New York, NY 10010*

**Reviewed on OpenReview:** *https://openreview.net/forum?id=p8gncJbMit*

## Abstract

We consider when a sparse nonnegative matrix $\mathsf{S}$ can be recovered, via an elementwise non-linearity, from a real-valued matrix $\mathsf{L}$ of significantly lower rank. Of particular interest is the setting where the positive elements of $\mathsf{S}$ encode the similarities of nearby points on a low dimensional manifold. The recovery can then be posed as a problem in manifold learning—in this case, how to learn a norm-preserving and neighborhood-preserving mapping of high dimensional inputs into a lower dimensional space. We describe an algorithm for this problem based on a generalized low-rank decomposition of sparse matrices. This decomposition has the interesting property that it can be encoded by a neural network with one layer of rectified linear units; since the algorithm discovers this encoding, it can be viewed as a layerwise primitive for deep learning. The algorithm regards the inputs $\boldsymbol{x}_i$ and $\boldsymbol{x}_j$ as similar whenever the cosine of the angle between them exceeds some threshold $\tau \in (0, 1)$. Given this threshold, the algorithm attempts to discover a mapping $\boldsymbol{x}_i \mapsto \boldsymbol{y}_i$ by matching the elements of two sparse matrices; in particular, it seeks a mapping for which $\mathsf{S} = \max(0, \mathsf{L})$, where $S_{ij} = \max(0, \boldsymbol{x}_i \cdot \boldsymbol{x}_j - \tau \|\boldsymbol{x}_i\| \|\boldsymbol{x}_j\|)$ and $L_{ij} = \boldsymbol{y}_i \cdot \boldsymbol{y}_j - \tau \|\boldsymbol{y}_i\| \|\boldsymbol{y}_j\|$. We apply the algorithm to data sets where vector magnitudes and small cosine distances have interpretable meanings (e.g., the brightness of an image, the similarity to other words). On these data sets, the algorithm discovers much lower dimensional representations that preserve these meanings.

## 1 Introduction

Many algorithms for high dimensional data analysis succeed by capitalizing on particular types of structure in large matrices. One important type of structure is sparsity; for example, when a matrix is sparse, with a large number of zero elements, it can be stored in a compressed format (Duff et al., 2017). Another type of structure is linear dependence; when a matrix is rank deficient, with one or more singular values equal to zero, it can be expressed as the product of two smaller matrices (Horn & Johnson, 2012). It is well known that neither one of these structures implies the other. For example, a very sparse matrix can be full rank (e.g., the identity matrix), and conversely, a very low rank matrix can be dense (e.g., the matrix of all ones).

It would be incorrect, however, to say that these types of structure are unrelated. One can find more subtle connections between rank and sparsity by looking beyond the canonical decompositions of linear algebra (Udell et al., 2016; Wright & Ma, 2021). One such connection was explored recently for sparse *nonnegative* matrices (Saul, 2022). In particular, let $\mathsf{S}$ denote such a matrix. Then one can find a real-valued matrix $\mathsf{L}$ of equal or lower rank such that

$$\mathsf{S} \approx \max(0, \mathsf{L}), \tag{1}$$

where the piecewise linear operation in eq. (1) is applied independently to all elements of $\mathsf{L}$. Note that a trivial result of this form is obtained by setting $\mathsf{L} = \mathsf{S}$ (where the approximation is exact). A more intriguing possibility emerges from the following observation: *the sparser the matrix $\mathsf{S}$, the larger the space that can be*

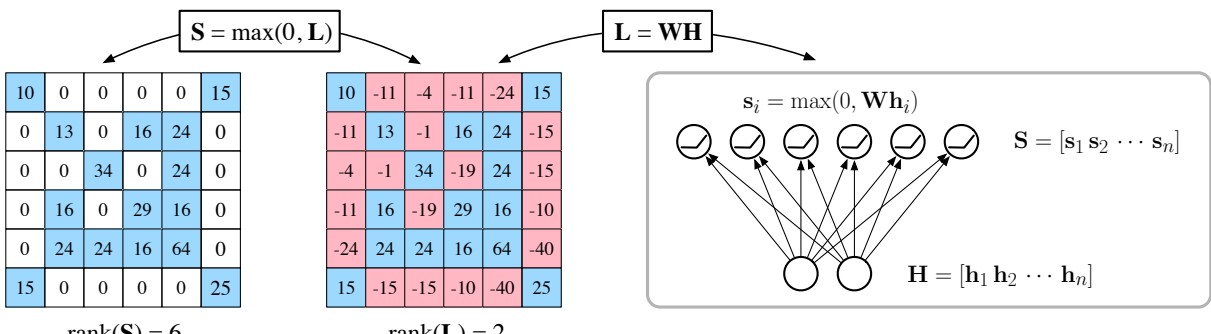

Figure 1: *Left:* A sparse nonnegative matrix **S** can be expressed via eq. (1) in terms of a matrix **L** of equal or lower rank. The matrices share the same positive elements (shown in **blue**), but the zeros of **S** may correspond to either zeros or negative values in **L** (shown in **red**). *Large gaps in rank may become possible when **S** is very sparse. Right:* The columns of **S** can be encoded by a neural network with a top layer of rectified linear units and a bottom layer of $d$ hidden units, where $d = \text{rank}(\textbf{L})$.

*explored to discover a matrix **L** of significantly lower rank.* The intuition behind this observation is illustrated in the left panels of Fig. 1. An even more striking example is the following. Let **I** denote the $n \times n$ identity matrix; then for all $n$, there exists a matrix **L** such that $\textbf{I} = \max(0, \textbf{L})$ and $\text{rank}(\textbf{L}) \leq 3$ (Saul, 2022).

The computation in eq. (1) is interesting for another reason—namely, that it can be implemented by a neural network with one layer of rectified linear units (ReLUs) (Mazumdar & Rawat, 2019a). To see this, suppose that the low-rank matrix in eq. (1) is factored as $\textbf{L} = \textbf{WH}$. Then we can view the columns of **H** as hidden units that encode the columns of **S**, and we can view the rows of **W** as weight vectors that reconstruct the rows of **S** from these encodings. This idea is illustrated in the right panel of Fig. 1.

Eq. (1) suggests many possibilities for high dimensional data analysis, especially for data that is naturally represented as a sparse nonnegative matrix (e.g., n-gram counts (Michel et al., 2011), affinities in a social network (Newman, 2003)). But eq. (1) by itself has several limitations, of which we briefly mention three:

1. It is not obvious how to interpret the nonlinear encodings of data that are obtained in this way. In particular, they suggest neither a subspace model of the data, as might be obtained by singular value decomposition (Eckart & Young, 1936; Deerwester et al., 1990; Turk & Pentland, 1991), nor a whole-parts decomposition, as might be obtained by nonnegative matrix factorization (Lee & Seung, 1999; Gillas, 2021) or a sparse dictionary method (Olshausen & Field, 1996; Chen et al., 1998; Mairal et al., 2010; Rubinstein et al., 2010). For many data sets, a geometrical picture is lacking.

2. It is not obvious how to apply the transformation in eq. (1) to data that is not sparse, even though many studies have shown such data can be modeled by neural networks with rectified linear units (Hinton & Ghahramani, 1997; Radford et al., 2016).

3. Based on the single-layer network in Fig. 1, one might be tempted to imagine eq. (1) as a greedy, *layerwise* strategy for unsupervised learning in deep networks (Hinton & Salakhutdinov, 2006; Hinton et al., 2006; Mazumdar & Rawat, 2019b). But Fig. 1 also reveals an immediate complication: unlike the original matrix **S**, the matrix **H** of hidden encodings is neither sparse nor nonnegative. Thus it is not obvious how to compose this method in a way that generalizes to deep architectures.

In this paper we describe a model, also based on eq. (1), that overcomes the first two of these limitations and suggests a way around the third. The model is designed to learn faithful low-dimensional representations of high-dimensional inputs in an unsupervised and highly interpretable manner. The inputs are not required to be sparse. This is the so-called problem of nonlinear dimensionality reduction, or manifold learning, and it has been widely studied (Tenenbaum et al., 2000; Roweis & Saul, 2000; Belkin & Niyogi, 2003; Coifman et al., 2005; Weinberger & Saul, 2006; van der Maaten & Hinton, 2008; McInnes et al., 2018; Agrawal et al.,

2021). Nevertheless, the model in this paper offers a novel solution, powered by eigenvector methods, based on the neural network and nonlinear matrix decomposition in Fig. 1. When applied to images, it is also designed to learn *intensity-equivariant* representations (Nair & Hinton, 2010; Hinton et al., 2011), and thus it differs from previous approaches that do not proceed with this goal in mind.

Most of this paper is devoted to describing this model and evaluating the simplest algorithm that it suggests to implement. But the main contribution of this paper is a conceptual one: it is to attach a geometrical picture to eq. (1). To begin, suppose that a data set of high-dimensional sensory inputs has been sampled from a low-dimensional perceptual manifold (Seung & Lee, 2000). The problem of manifold learning can be viewed, at a high level, as one of translating between the two different types of matrices that arise in this setting. On one hand, there are sparse matrices, whose nonzero elements reveal at a glance which high dimensional inputs are to be regarded as similar or nearby on the manifold (Sengupta et al., 2018; Chen et al., 2018); on the other hand, there are low-rank matrices whose nonzero eigenvalues reveal at a glance the dimensionality of some Euclidean space in which the manifold can be embedded (Tenenbaum et al., 2000; Weinberger & Saul, 2006). It is in this sense that certain sparse and low-rank matrices can be viewed as "two sides" of the same manifold, each revealing different aspects of its structure. The main contribution of this paper is the following: *when properly formulated, these different matrices—one sparse, one low-rank—are mathematically connected by eq. (1).* This connection is both surprisingly simple, involving only a piecewise linearity, and also deeply familiar, involving the same piecewise linearity that is most widely used in applications of deep learning. One might even say that the goal of this paper is to reveal a geometrical connection between sparse and low-rank matrices that was hiding in plain sight.

The organization of this paper is as follows. In section 2, we develop a model for nonlinear dimensionality reduction in which input similarities are encoded by a sparse matrix $\mathbf{S}$ of thresholded inner products. Given this sparse matrix $\mathbf{S}$, the model's embeddings are obtained by solving a sequence of eigenvalue problems for the corresponding low-rank matrix $\mathbf{L}$ in eq. (1). In section 3, we describe one way to extend this model to larger data sets where it is too expensive to construct and/or solve these eigenvalue problems directly. This extension requires only the solution of a sparse least-squares problem. In both of these sections, we also apply the model to images of handwritten digits (LeCun et al., 1998) and pre-trained word vectors (Mikolov et al., 2018). Finally, in section 4, we conclude by discussing multiple avenues for future work.

## 2 Thresholded similarity matching

We begin by stating the problem that will ultimately lead us back to eq. (1). Let $\mathcal{X} = \{\boldsymbol{x}_1, \boldsymbol{x}_2, \ldots, \boldsymbol{x}_n\}$ denote a data set of high-dimensional inputs in $\Re^D$. Our goal is to discover a corresponding set of outputs $\mathcal{Y} = \{\boldsymbol{y}_1, \boldsymbol{y}_2, \ldots, \boldsymbol{y}_n\}$ in $\Re^d$ that provide a faithful but much lower-dimensional representation (or *embedding*) of the data (with $d \ll D$). Section 2.1 defines what we mean by faithful in this context, section 2.2 describes an algorithm to compute the embedding, section 2.3 presents the results of this algorithm on two data sets of interest, and section 2.4 relates our approach to previous work on this problem.

### 2.1 Model

What does it mean for an embedding to be a faithful representation? We start from a familiar place, measuring similarities between vectors via inner products. As shorthand, let $\cos(\boldsymbol{x}, \boldsymbol{x}') = \frac{\boldsymbol{x} \cdot \boldsymbol{x}'}{\|\boldsymbol{x}\| \|\boldsymbol{x}'\|}$ denote the cosine of the angle between the vectors $\boldsymbol{x}$ and $\boldsymbol{x}'$. Then we define an embedding to be a $\tau$-*faithful representation* of the data (with $0 < \tau < 1$) if it meets the following three conditions:

    (i) Magnitudes are preserved:   $\|\boldsymbol{y}_i\| = \|\boldsymbol{x}_i\|$ for all $i$.
    (ii) Small angles are preserved:   $\cos(\boldsymbol{y}_i, \boldsymbol{y}_j) = \cos(\boldsymbol{x}_i, \boldsymbol{x}_j)$ whenever $\cos(\boldsymbol{x}_i, \boldsymbol{x}_j) > \tau$.
    (iii) Large angles remain large:   $\cos(\boldsymbol{y}_i, \boldsymbol{y}_j) \leq \tau$ whenever $\cos(\boldsymbol{x}_i, \boldsymbol{x}_j) \leq \tau$.

Likewise we shall say (informally) that an embedding is an approximately $\tau$-faithful representation if these conditions are approximately met. Note that equalities appear in the first two conditions, but not the third. In particular, large angles do not need to be preserved; they only need to remain larger than $\cos^{-1}(\tau)$.

We emphasize the following point: it is by design that these conditions refer to magnitudes and angles as opposed to pairwise distances (or proximities), as is more commonly done in studies of nonlinear dimensionality reduction (Tenenbaum et al., 2000; Belkin & Niyogi, 2003; Weinberger & Saul, 2006; van der Maaten & Hinton, 2008; McInnes et al., 2018; Agrawal et al., 2021). The first condition is particularly apt for data sets in which the magnitude of inputs convey meaningful information in their own right. In this paper, for example, we apply the model to data sets of images and word vectors. Note that the magnitude of a pixel vector measures the overall brightness of an image, while the magnitude of a word vector reflects a word's specificity (Schakel & Wilson, 2015) (i.e., whether it can be used in one or many different contexts). Thus the origins in these spaces have a definite meaning (e.g., a blank image, a nondescript word), and the first condition requires that the origin in the input space is mapped uniquely to the origin in the output space, thus preserving this meaning. In fact, the second condition requires further that parallel inputs are mapped to parallel outputs. This is necessary to learn intensity-equivariant mappings (Nair & Hinton, 2010; Hinton et al., 2011), as is desirable in sensory domains where the perceived strength of a signal is coded by its magnitude (e.g., the brightness of images, the volume of sounds). It should also be noted that these conditions are *not* appropriate for all data sets. For example, they would thwart the "unfoldings" of many popular but artificial data sets (e.g., Swiss rolls) where the magnitudes of inputs do not have this interpretation.

The name of our model is derived from the following observation. Note how *all three* of the above conditions are compactly captured by the single equation:

$$\max(0, \boldsymbol{x}_i \cdot \boldsymbol{x}_j - \tau \|\boldsymbol{x}_i\| \|\boldsymbol{x}_j\|) \;=\; \max(0, \boldsymbol{y}_i \cdot \boldsymbol{y}_j - \tau \|\boldsymbol{y}_i\| \|\boldsymbol{y}_j\|). \tag{2}$$

Now suppose in this equation that the threshold $\tau \in (0, 1)$ exceeds the cosine of the angle between most pairs of inputs in $\mathcal{X}$. Then the left side will define a highly sparse matrix, and a $\tau$-faithful embedding will only be obtained if the same thresholded matrix is produced on the right side by the outputs in $\mathcal{Y}$. Next consider the *dense* matrix $\boldsymbol{y}_i \cdot \boldsymbol{y}_j - \tau \|\boldsymbol{y}_i\| \|\boldsymbol{y}_j\|$ on the right side of eq. (2). If the outputs $\{\boldsymbol{y}_i\}_{i=1}^n$ live in $\Re^d$, then the rank of this matrix can be at most $d+1$. Hence if there does exist a $\tau$-faithful embedding into $\Re^d$, then there also exists a matrix of rank at most $d+1$ from which the sparse matrix on the left side can be recovered. These are the essential ideas behind our approach, which we call *Thresholded Similarity Matching* (TSM).

The sparsity of the matrices in eq. (2) is governed by the threshold $\tau$, which must be chosen manually. Given a data set $\mathcal{X}$, a reasonable value for $\tau$ can be determined by examining the histograms of angles between nearest neighbors (using cosine distance) versus other pairs of inputs; the goal is to identify a threshold above which inputs can be meaningfully regarded as similar. Some of these histograms are shown in Fig. 2 for two data sets—one of images of MNIST handwritten digits (LeCun et al., 1998), and one of pre-trained word vectors from a FastText embedding (Mikolov et al., 2018). (More details on these data sets are given in section 2.3.) The figure highlights particular choices of $\tau$ for these data sets; for these choices, the matrix on the left side of eq. (2) is 99% sparse but has nonzero entries for nearly all pairs of one-nearest neighbors (1NN). Put another way, $\tau$-faithful embeddings with these values of $\tau$ will preserve the angles between the top one-percent of closely aligned inputs. Note in these embeddings that angles are preserved not only between 1NNs, but also between all pairs of inputs whose cosines exceed the threshold $\tau$.

## 2.2 Algorithm

Next we describe how to compute an approximately $\tau$-faithful embedding of some specified dimensionality $d$. Our algorithm proceeds in three steps.

### Step 1. Compute the sparse matrix S.

To begin, we identify the left side of eq. (2) with the sparse matrix **S** in eq. (1):

$$S_{ij} = \max(0, \boldsymbol{x}_i \cdot \boldsymbol{x}_j - \tau \|\boldsymbol{x}_i\| \|\boldsymbol{x}_j\|). \tag{3}$$

The algorithm starts simply by computing this matrix. There is only one possible complication in this step; it arises when a row or column of **S** contains only a single nonzero element along the diagonal. This situation occurs when an input $\boldsymbol{x}_i$ (typically an outlier) is not within an angle of $\cos^{-1}(\tau)$ from any other inputs

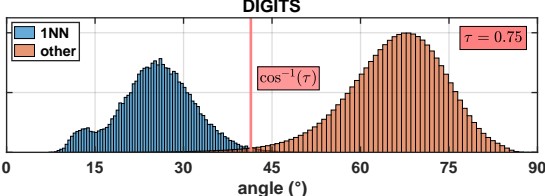 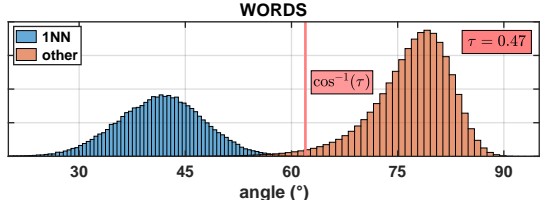

Figure 2: Histograms of angles between inputs that are one-nearest neighbors (1NNs) in **blue** versus those that are not in **orange**. The histograms are separately normalized as there are many fewer pairs of inputs in the first category. The histograms were used to set the threshold, $\tau$, that identifies pairs of similar inputs. We set $\tau = 0.75$ for images of handwritten digits (*left*) and $\tau = 0.47$ for word vectors (*right*); the angles corresponding to these values are indicated by the **red** vertical lines. For these choices, the matrix on the left side of eq. (2) was 99% sparse. Note that the overlap between histograms is small but nonzero, and for both data sets there remain a large number of kNN (with $k > 1$ but small) that are regarded as similar.

besides itself. In this case, the matrix **S** may not provide sufficient information to co-locate the output $\boldsymbol{y}_i$ in the low-dimensional embedding (indicating only that it should be far from every other ouput). In practice, when this situation occurs, we augment the data set $\mathcal{X}$ to include *virtual* inputs that interpolate between $\boldsymbol{x}_i$ and its nearest neighbor and that differ by angles less than $\cos^{-1}(\tau)$. Note that if this situation[1] occurs for more than a small fraction of inputs, then it suggests to lower the threshold $\tau$ in eqs. (2–3).

**Step 2. Compute the low-rank matrix L.**

Next the algorithm attempts to find a low-rank matrix **L** from which the sparse matrix **S** in eq. (3) can be recovered via eq. (1). In our experience, the algorithm works best when the rank of **L** is equated to the target dimensionality ($d$) of the desired embedding, but higher choices are also possible.

For this step we adapt an alternating minimization procedure that was recently proposed for this problem (Saul, 2022). This procedure alternates between two updates—one for the matrix **L**, and another for an auxiliary matrix **Z** of the same size. The updates attempt to solve the constrained optimization:

$$\min_{\mathbf{L},\mathbf{Z}} \left\| \mathbf{L} - \mathbf{Z} \right\|_F^2 \quad \text{such that} \quad \begin{cases} d = \text{rank}(\mathbf{L}), \\ \mathbf{S} = \max(0, \mathbf{Z}). \end{cases} \tag{4}$$

Note that the objective function in eq. (4) is bounded below by zero, and it only attains this minimum value when $\mathbf{S} = \max(0, \mathbf{L})$ and **L** is a matrix of rank $d$. Though not explicitly indicated in eq. (4), both **L** and **Z** are understood to be symmetric matrices (like **S**).

The alternating updates for this optimization take an especially simple form. When the matrix **L** is fixed, the objective function is minimized by setting

$$Z_{ij} = \begin{cases} S_{ij} & \text{if } S_{ij} > 0, \\ \min(0, L_{ij}) & \text{otherwise.} \end{cases} \tag{5}$$

Likewise, when the matrix **Z** is fixed, the objective function is minimized by setting **L** to the best low-rank approximation of **Z** as measured by the Frobenius norm in eq. (4). Thus the update for **L** takes the form:

$$\mathbf{L} = \arg\min_{\mathbf{M}} \| \mathbf{M} - \mathbf{Z} \|_F^2 \quad \text{such that} \quad \text{rank}(\mathbf{M}) = d. \tag{6}$$

Since **Z** is symmetric (though generally not positive definite), the solution to eq. (6) is obtained by setting $\mathbf{L} = \mathbf{Q}\mathbf{D}\mathbf{Q}^\top$, where **D** is the $d \times d$ diagonal matrix whose diagonal elements specify the largest $d$ eigenvalues (in magnitude) of **Z** and **Q** is the corresponding $n \times d$ matrix of normalized eigenvectors.

---

[1] All graph-based approaches to nonlinear dimensionality reduction are plagued to some extent by outliers. Other alternatives (not considered here) are to analyze only the largest connected component of inputs (Tenenbaum et al., 2000) or to construct a neighborhood graph via minimum spanning trees (Zemel & Carreira-Perpiñán, 2004), thus ensuring that it is connected.

The optimization in eq. (4) is not convex (due to the constraints on **L** and **Z**), and hence this procedure is not guaranteed to converge to a global minimum. By construction, however, each of the updates in eqs. (5–6) reduces the value of the objective function (except at stationary points), and thus this property provides a guarantee of monotonic improvement. In practice, the updates can be alternated as long as the objective function is decreasing at each iteration by some small amount. The algorithm works on the same general principle as other alternating minimizations, such as for Expectation-Maximization (Dempster et al., 1977; Meng & van Dyk, 1997; McLachlan & Krishnan, 2008) and nonnegative matrix factorization (Lee & Seung, 1999; Gillas, 2021), and thus similar empirical strategies can be used to test for its convergence. We refer the reader to appendix A for further details of implementation; these details include the choice of initialization, an additional linear constraint on the elements of **Z**, and the use of a momentum parameter, all of which lead to improved solutions and/or faster convergence.

**Step 3. Compute the embedding $\mathcal{Y}$.**

The last step of the algorithm is to recover the $d$-dimensional outputs $\{\boldsymbol{y}_1, \ldots, \boldsymbol{y}_n\}$. To do so, we identify the matrix argument on the right side of eq. (2) with the low-rank matrix **L** computed in the previous step:

$$L_{ij} = \boldsymbol{y}_i \cdot \boldsymbol{y}_j - \tau \|\boldsymbol{y}_i\| \|\boldsymbol{y}_j\|. \tag{7}$$

The problem now is to recover the outputs $\boldsymbol{y}_i \in \Re^d$ from the known elements of **L**. A similar problem arises in multidimensional scaling (Cox & Cox, 2000), where one recovers $n$ vectors (centered on the origin) from an $n \times n$ matrix of pairwise squared distances (e.g., $\|\boldsymbol{y}_i - \boldsymbol{y}_j\|^2$). The problem here is analogous, except that we are recovering the outputs from the norms and inner products on the right side of eq. (7); moreover, the outputs in our problem are not required to be centered.

We first derive an exact solution to this problem under the assumption that one does, in fact, exist. Assuming that eq. (7) holds for some outputs $\boldsymbol{y}_i \in \Re^d$, then it follows from the diagonal elements that $L_{ii} = (1-\tau)\|\boldsymbol{y}_i\|^2$, or equivalently, $\|\boldsymbol{y}_i\| = \sqrt{L_{ii}}(1-\tau)^{-1/2}$. Next we substitute this result into the right side of eq. (7). This yields an expression for the Gram matrix **G** whose elements $G_{ij} = \boldsymbol{y}_i \cdot \boldsymbol{y}_j$ store the inner products of the outputs. In particular, we have:

$$G_{ij} = L_{ij} + \frac{\tau}{1-\tau}\sqrt{L_{ii}L_{jj}} \tag{8}$$

The outputs are then determined (up to an arbitrary rotation) by the elements of this Gram matrix, thus completing the solution.

The above solution was based on an assumption—namely, that there exist outputs whose inner products are perfectly consistent with the matrix **L** computed in the previous step of the algorithm. When this is not the case, we compute an approximate solution that is optimal in a least-squares sense (Cox & Cox, 2000; Tenenbaum et al., 2000). Let $\{\lambda_\alpha\}_{\alpha=1}^d$ denote the $d$ largest eigenvalues of the matrix **G**, and let $\{\boldsymbol{e}_\alpha\}_{\alpha=1}^d$ denote the corresponding eigenvectors. Then an approximate solution to eq. (7) is given by

$$y_{\alpha i} = \sqrt{\max(0, \lambda_\alpha)}\, e_{i\alpha}. \tag{9}$$

When the matrix **G** from eq. (8) is positive semidefinite, this approximation computes the outputs in $\Re^d$ (up to a global rotation) that minimize the error $\sum_{ij}(G_{ij} - \boldsymbol{y}_i \cdot \boldsymbol{y}_j)^2$. When this is not the case, the approximation proceeds by ignoring the negative eigenvalues of **G**. (Large or multiple negative eigenvalues suggest that one should choose a higher value for the target dimensionality $d$.)

**Practical considerations.** The algorithm proceeds from choices of the threshold $\tau$ in step 1 and the target dimensionality $d$ in step 2. It should be clear that these choices are interrelated. The smaller the value of $\tau$, the larger the number of inner products that the algorithm attempts to preserve in steps 1 and 2. Likewise, the smaller the value of $d$, the less flexibility there is to discover a similarity-preserving embedding in steps 2 and 3. The main question is the following: is there a meaningful regime for these choices in which (a) the optimization in eq. (4) is not eviscerated by local minima, and (b) the low-rank matrix **L** satisfying $\mathbf{S} \approx \max(0, \mathbf{L})$ yields a sensible Gram matrix via eq. (8)? This question can be answered to some degree by empirical evaluations, to which we turn next.

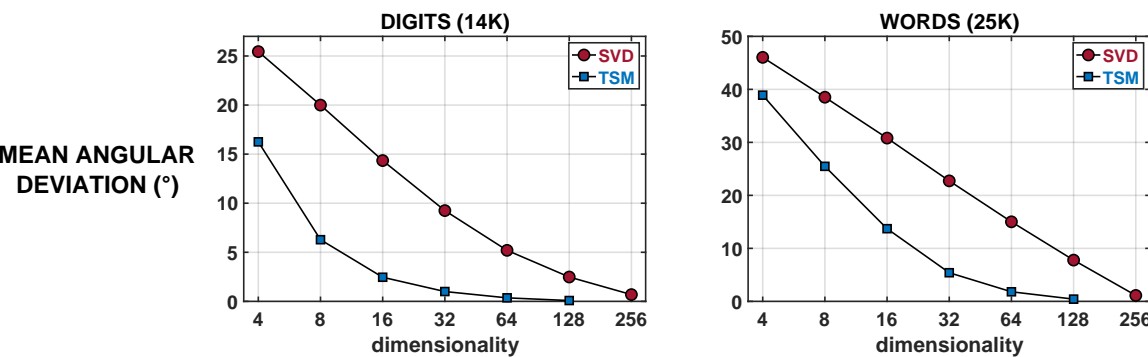

Figure 3: Mean angular deviations in eq. (11) for the embeddings from **TSM** and **SVD** of 14K digit images (*left*) and 25K word vectors (*right*), as a function of their dimensionality.

## 2.3 Evaluation and results

**Data sets.** We evaluated the algorithm of the previous section on two data sets where it is common to measure similarity using cosine distances:

1. *Handwritten digit images*
   The first data set was the MNIST data set of grayscale handwritten digits (LeCun et al., 1998). In this data set, each image is encoded by a 784-dimensional vector of pixel values. The data set contains a total of 70K images, which we divided into five non-overlapping folds of 14K images. We ran the algorithm separately on each of these folds.

2. *Pre-trained word vectors*
   The second data set consisted of a subset of pre-trained 300-dimensional word vectors from a FastText embedding of 1M English tokens (Mikolov et al., 2018). For the experiments in this section, we ran the algorithm on a subset of approximately 25K word vectors that were selected to obtain a sub-vocabulary of mostly contentful words. This sub-vocabulary included the leading 18K words of the FastText embedding that are valid tournament plays in the game of Scrabble[2], as well as all 905 unique words in the Semantic-Syntactic Word Relationship test set for analogy completion (Mikolov et al., 2013a;b). The latter were included so that we could use analogy completion to assess different low-dimensional embeddings of word-vectors; to do this, it was necessary to supplement our sub-vocabulary with (for instance) the many proper nouns, such as state capitals, that appear in these analogies but are not valid plays in Scrabble. Finally, we also included the 15-nearest neighbors (in cosine distance) for each of these 905 word-vectors so that they did not appear as isolated instances in our data set but were linked to other words with similar meanings. This brought the total sub-vocabulary to 25124 words.

The data sets were restricted in size for these initial evaluations because the updates in eqs. (5–6) work with dense $n \times n$ matrices (namely, $\mathbf{Z}$ and $\mathbf{L}$), and these matrices must fit comfortably into memory. We evaluate a more memory-efficient way to handle larger data sets in section 3.

**Baselines.** To obtain baseline embeddings, we performed a truncated singular value decomposition (SVD) of the above data sets; the inputs were then projected into the lower-dimensional subspace spanned by the top left singular vectors. More concretely this was done as follows. Let $\mathbf{X} = [\boldsymbol{x}_1 \, \boldsymbol{x}_2 \, \ldots \, \boldsymbol{x}_n]$ denote the matrix of high dimensional inputs, and let $\mathbf{U}$ denote the matrix whose columns store the top $d$ eigenvectors of $\mathbf{X}\mathbf{X}^\top$. We computed the linear embedding $\boldsymbol{x}_i \to \boldsymbol{v}_i$ via the orthogonal projections

$$\boldsymbol{v}_i = \mathbf{U}^\top \boldsymbol{x}_i. \tag{10}$$

---

[2]An official tournament word list is available at `https://www.wordgamedictionary.com/twl06/download/twl06.txt`.

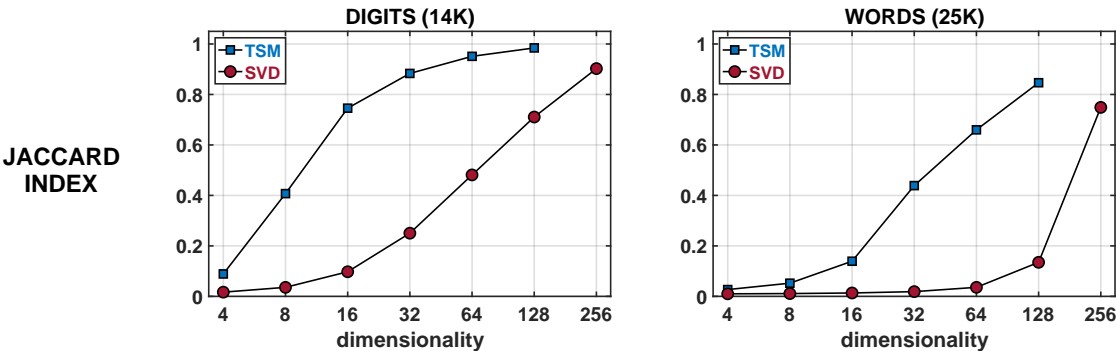

Figure 4: Jaccard indices in eq. (12) for the embeddings from **TSM** and **SVD** of 14K digit images (*left*) and 25K word vectors (*right*), as a function of their dimensionality.

Since these projections are orthogonal, the outputs in these embeddings always have smaller norms than their corresponding inputs: $\|\boldsymbol{v}_i\| \leq \|\boldsymbol{x}_i\|$. As discussed next, however, we chose metrics for evaluation that did not penalize the baseline embeddings for this property. These linear embeddings are optimal in a least-squares sense: if $\mathbf{P} = \mathbf{U}\mathbf{U}^\top$, then $\mathbf{P}$ is also the projection matrix of rank $d$ that minimizes the error $\|\mathbf{X} - \mathbf{P}\mathbf{X}\|_F^2$.

**Metrics.** We computed two fairly intuitive metrics to evaluate the quality of embeddings $\boldsymbol{x}_i \to \boldsymbol{y}_i$. As a useful shorthand, let $\Omega_x = \{(i,j) \,|\, S_{ij} > 0\}$ denote the set of indices where the matrix $\mathbf{S}$ in eq. (3) has positive (i.e., nonzero) elements, and let $|\Omega_x|$ denote the total number of these positive elements. The first metric we computed was the mean angular deviation

$$\Delta = \frac{1}{|\Omega_x|} \sum_{ij \in \Omega_x} \left| \cos^{-1}\left( \frac{\boldsymbol{x}_i \cdot \boldsymbol{x}_j}{\|\boldsymbol{x}_i\|\|\boldsymbol{x}_j\|} \right) - \cos^{-1}\left( \frac{\boldsymbol{y}_i \cdot \boldsymbol{y}_j}{\|\boldsymbol{y}_i\|\|\boldsymbol{y}_j\|} \right) \right|. \tag{11}$$

Note that the sum in eq. (11) is restricted to those pairs of inputs whose angles the algorithm is trying to preserve. As another useful shorthand, let $\Omega_y = \{(i,j) \,|\, \boldsymbol{y}_i \cdot \boldsymbol{y}_j - \tau\|\boldsymbol{y}_i\|\|\boldsymbol{y}_j\| > 0\}$ denote the set of indices where the matrix on the right side of eq. (2) has positive elements. The second metric we computed was the Jaccard index (Jaccard, 1901; Levandowsky & Winter, 1971):

$$J(\Omega_x, \Omega_y) = \frac{|\Omega_x \cap \Omega_y|}{|\Omega_x \cup \Omega_y|} \tag{12}$$

The Jaccard index lies between 0 and 1; it is equal to 1 when exactly the same pairs of inputs and outputs have angles less than $\cos^{-1}(\tau)$, and it is equal to 0 when none of the same pairs share this property.

**Comparisons.** Figures 3 and 4 compare these metrics for the embeddings[3] obtained from TSM and SVD. The embeddings were computed over a wide range of target dimensionalities for the data sets of 14K digit images[4] and 25K word vectors. The results are similar for both data sets: over a wide range of dimensionalities, the mean angular deviation is significantly lower for the TSM embeddings, while the Jaccard index is significantly higher. The results confirm the intuition suggested by Fig. 1; the nonlinear matrix decomposition in eq. (3) is capable of deriving faithful low-dimensional embeddings by capitalizing on the sparsity of the matrix $\mathbf{S}$.

**Analogy completion.** Analogies have been used to test the meaningfulness[5] of adding and subtracting word-vectors in different embeddings (Mikolov et al., 2013a;b). Consider, for example, the following analogy:

---

[3]Appendix B presents an additional but less straightforward comparison to results from the Isomap algorithm.

[4]For the digit images, the results in the figures were obtained by averaging over the five different folds of the full MNIST data set; the variation across folds, not shown in the figure, was generally smaller than the marker sizes.

[5]It is important to safeguard against stereotypes and implicit biases when word-vectors are used for real-world applications (Bolukbasi et al., 2016; Gonen & Goldberg, 2019). This is especially true when such biases might be inadvertently exacerbated by dimensionality reduction.

Table 1: Examples of analogies in different semantic and syntactic categories. These are also examples where the lower dimensional word-vectors from TSM (with target dimensionality $d=32$) predict the correct answer (shown in **blue**), while those from SVD predict an incorrect answer (shown in **red**).

| Type | Category | Prompt | Query | TSM | SVD |
|---|---|---|---|---|---|
| **Semantic** | common capitals | (Athens, Greece) | Baghdad | **Iraq** | **Afghanistan** |
| | other capitals | (Abuja, Nigeria) | Amman | **Jordan** | **Morocco** |
| | currencies | (Algeria, dinar) | Cambodia | **riel** | **rupee** |
| | city-state | (Chicago, Illinois) | Houston | **Texas** | **Mississippi** |
| | family | (boy, girl) | father | **mother** | **daughter** |
| **Syntactic** | adjective-adverb | (amazing, amazingly) | calm | **calmly** | **gently** |
| | opposites | (aware, unaware) | honest | **dishonest** | **cynical** |
| | comparisons | (bad, worse) | cold | **colder** | **damp** |
| | superlatives | (bad, worst) | cool | **coolest** | **awesome** |
| | participles | (code, coding) | jump | **jumping** | **leap** |
| | nationalities | (Albania, Albanian) | Austria | **Austrian** | **Hungarian** |
| | past tense | (dancing, danced) | describing | **described** | **represented** |
| | plural nouns | (banana, bananas) | goat | **goats** | **cows** |
| | plural verbs | (decrease, decreases) | estimate | **estimates** | **yields** |

"boy" is to "girl" as "father" is to "mother." Given an embedding of word-vectors, one can attempt to complete the analogy automatically by computing the translated vector

$$t_{\text{analogy}} = \texttt{vec}(\text{"father"}) + \left[ \texttt{vec}(\text{"girl"}) - \texttt{vec}(\text{"boy"}) \right], \tag{13}$$

then finding the word-vector (other than "father" or "girl") nearest to $t_{\text{analogy}}$ in cosine distance. We compared these sorts of completions (limited to our sub-vocabulary of 25K words) from the original FastText embedding and the lower-dimensional embeddings obtained from TSM and SVD. The completions were evaluated on the Semantic-Syntactic Word Relationship data set (Mikolov et al., 2013a;b), which consists of 19558 analogies from 14 different categories; of these 14 categories, 5 are based on semantic relationships, and 9 are based on syntactic relationships. Table 1 shows an example from each category, and Fig. 5 plots the overall accuracy rates for analogy completion, broken down by type. For semantic relationships, we see that the lower-dimensional embeddings from TSM approach the accuracy of the FastText embedding much more quickly than those from SVD. For syntactic relationships, the embeddings from TSM are also better at small dimensionalities, though the difference is less pronounced. A qualitative sense of these differences can be seen from the examples that appear in Table 1. These examples were selected from analogies that only the embedding from TSM completed correctly in $d=32$ dimensions.

## 2.4 Related work

The algorithm for TSM shares a familiar three-step recipe with many other eigenvector-based approaches to nonlinear dimensionality reduction. In a nutshell, the first of these steps computes a sparse matrix $\mathsf{S}$ from pairs of nearby inputs, the second constructs a related matrix of the same size (e.g., of graph distances (Tenenbaum et al., 2000), of linear reconstruction weights (Roweis & Saul, 2000), of graph Laplacian elements (Belkin & Niyogi, 2003), of "unfolded" inner products (Weinberger & Saul, 2006)), and the final step obtains an embedding from the top or bottom eigenvectors of this second matrix. *Within this framework, however, there remain several important differences.* First, previous approaches have not attempted to preserve the meaningfulness of norms and angles, and most of them (for instance) do not map the origin in the input space to the origin in the output space. Second, the algorithm for TSM enforces a low-rank constraint in its alternating minimization, whereas other algorithms in this framework typically obtain a nested set of embeddings that are not optimized for a particular target dimensionality. Third, and most importantly, TSM is based on the connection in eq. (1) between the matrices $\mathsf{S}$ and $\mathsf{L}$, and thus it provides a bridge between these eigenvector-based approaches and other neural-network models of unsupervised learning.

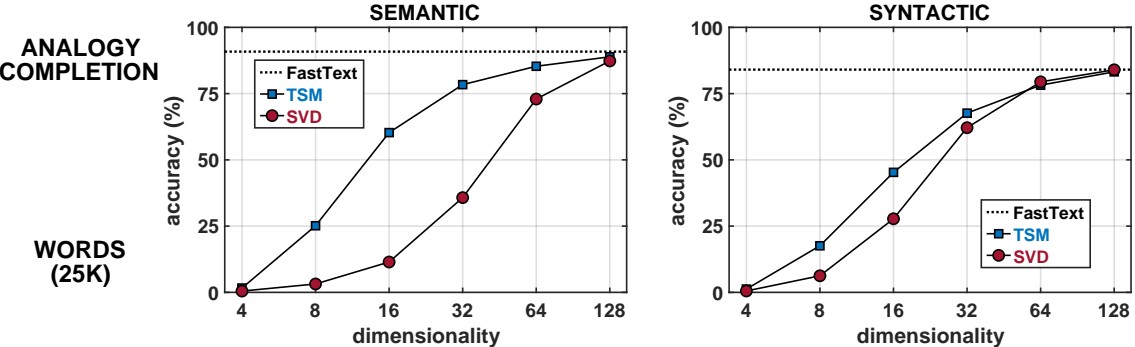

Figure 5: Analogy completion rates for the embeddings from **TSM** and **SVD** of 25K word vectors, as a function of their dimensionality, for semantic (*left*) and syntactic (*right*) relationships.

Another line of work in dimensionality reduction has focused on embeddings for visualization. Well-known algorithms for this purpose include t-SNE (van der Maaten & Hinton, 2008), UMAP (McInnes et al., 2018), and minimum distortion embedding (Agrawal et al., 2021). These algorithms search for embeddings that balance attractive and repulsive forces between their low-dimensional outputs. In general, there are many possible choices for these forces, and they tend to be validated by the visualizations they produce. As such, they have been less explored for higher-dimensional embeddings. Unlike TSM, they rely on gradient-based optimizations. With clever approximations (van der Maaten, 2014; Linderman et al., 2019) and sampling-based methods (McInnes et al., 2018; Agrawal et al., 2021), they have been scaled to very large data sets.

TSM is based on a nonlinear decomposition originally proposed for sparse nonnegative matrices (Saul, 2022). It builds on previous work by developing a geometrical picture for this decomposition. With this picture, eq. (1) can also be applied to data that is not intrinsically sparse; this is done by using the sparse matrix **S** to store thresholded similarities and the low-rank matrix **L** to recover a low-dimensional embedding.. TSM also fits within a long line of work on generalized low-rank models that search for low dimensional structure in nonnegative or binary data (Lee & Sompolinsky, 1999; Collins et al., 2002; Hoff, 2005; Udell et al., 2016).

Finally, we note that our approach was motivated in large part by a recent model of nonnegative similarity matching in biologically-inspired neural networks (Sengupta et al., 2018). The authors of this work observed that localized receptive fields emerge when neural networks are trained to optimize a similarity-matching objective function. They also showed—both numerically and analytically—that the receptive fields in these networks learn to tile the low-dimensional manifolds that are populated by their sensory inputs. The present work began with the realization that the responses of these receptive fields could be collected to form a sparse matrix **S**, such as the one that appears in eq. (1). The threshold $\tau$ in TSM also plays a role analogous to the similarity threshold in their work.

## 3 Locally linear extension

In this section we describe one way to extend the embeddings from TSM to larger data sets. The approach is based on familiar ideas from graph-based semi-supervised learning (Zhu et al., 2003) and large-scale manifold learning (Weinberger et al., 2005; Vladymyrov & Carreira-Perpiñán, 2013). With this approach, we were able obtain results on larger data sets with only slightly more computation than the experiments in the previous section.

### 3.1 Algorithm

We begin by reframing the problem. We suppose that the method of the previous section has been used to obtain an embedding of $m$ inputs, and now we wish to extend the embedding to an additional $n-m$ inputs. We refer to the original $m$ outputs as *landmarks*, and we propose a two-step algorithm to compute a *locally linear extension* (LLX) of the smaller embedding.

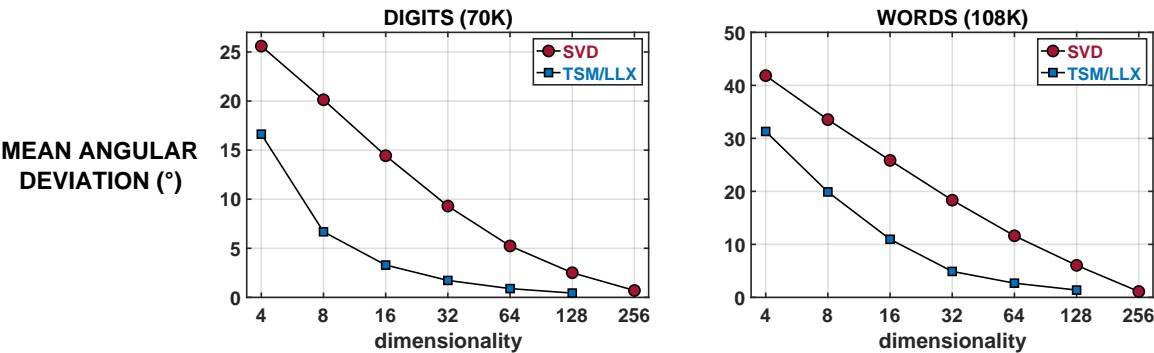

Figure 6: Mean angular deviations in eq. (11) for the embeddings from **TSM/LLX** and **SVD** of 70K digit images (*left*) and 108K word vectors (*right*), as a function of their dimensionality.

**Step 1. Compute a sparse matrix W of reconstruction weights.** This step is similar to the first step of the algorithm known as locally linear embedding (LLE) (Roweis & Saul, 2000). For each input $\boldsymbol{x}_i$, we compute a set of $k$ weights that reconstruct the input (approximately) from its $k$-nearest neighbors. Let $\mathcal{K}_i$ denote the set of $k$-nearest neighbors for input $\boldsymbol{x}_i$. Then these weights can be computed by minimizing the regularized sum of locally linear reconstruction errors,

$$E_{\mathcal{X}}(\mathbf{W}) = \left\| \boldsymbol{x}_i - \sum_{j \in \mathcal{K}_i} W_{ij}\boldsymbol{x}_j \right\|^2 \; + \; \varepsilon \sum_{j \in \mathcal{K}_i} \|\boldsymbol{x}_j\|^2 \, W_{ij}^2, \tag{14}$$

where $\epsilon > 0$ is a small value to avoid ill-posed problems. There are two minor but important differences with the original algorithm for LLE: the first is that nearest neighbors are selected according to cosine distance, and the second is that the rows of **W** are *not* constrained to sum to one. Both differences arise from the desire to preserve the meaningfulness of norms and small cosine distances.

**Step 2. Extend the embedding.** The embedding is extended by assuming that nearby outputs should be linearly related in the same way as nearby inputs (Weinberger et al., 2005; Vladymyrov & Carreira-Perpiñán, 2013). We therefore proceed by minimizing the analogous reconstruction error

$$E_{\mathbf{W}}(\mathcal{Y}) = \left\| \boldsymbol{y}_i - \sum_{j} W_{ij}\boldsymbol{y}_j \right\|^2 \tag{15}$$

over the locations of the $n-m$ unknown outputs. To emphasize, the $m$ landmarks in eq. (15) are fixed to their known values, while the weights $W_{ij}$ are determined previously by the values that minimize eq. (14). The minimization of eq. (15) reduces to $d$ independent least-squares problems for the coordinates of the $n-m$ unknown outputs. Most importantly, because the matrix **W** is sparse, these problems can be solved very efficiently using preconditioned conjugate gradients methods (Barrett et al., 1994). Further details of this approach can be found in the appendix.

### 3.2 Results

We experimented with this algorithm, which we denote by TSM/LLX, on the full data set of MNIST handwritten digits and a larger data set of word vectors. For both data sets, we used the algorithm to extend an embedding of $m$ outputs (computed by TSM) to an embedding of $n = 5m$ outputs (computed by LLX). For the digits, we used $m = 14000$ and $n = 70000$ images, and for the text embedding, we used $m = 21534$ and $n = 107670$ word vectors. The vocabulary in this larger data set of word-vectors was expanded from the previous one to include all valid Scrabble plays that were among the leading one million tokens of the FastText embedding. For these experiments, we set $k = 4d$ in eq. (14) and $\epsilon = 10^{-4}$ in eq. (15).

Figures 6 and 7 show the results of these experiments, comparing the mean angular deviations and Jaccard indices of TSM/LLX embeddings versus those of SVD. These results were averaged over the five non-

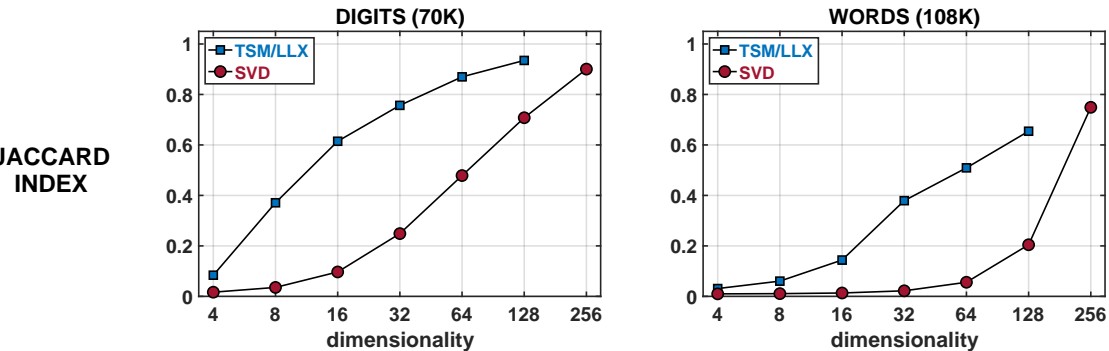

Figure 7: Jaccard indices in eq. (12) for the embeddings from **TSM/LLX** and **SVD** of 70K digit images (*left*) and 108K word vectors (*right*), as a function of their dimensionality.

overlapping folds of each data set (where a particular fold specified which $m$ of $n$ examples were chosen as landmarks). Here again, we see that the TSM/LLX embeddings are much better as evaluated by these metrics. As to be expected, these TSM/LLX embeddings have somewhat smaller gains than the TSM embeddings in Figs. 3 and 4 which do not involve an additional layer of approximation. Nevertheless, the gains are still quite significant. Though not shown in the figures, we also observed in these experiments that the results did not vary much across different folds.

## 4 Conclusion

In this paper we have explored a geometrical connection between sparse and low-rank matrices. The connection emerged from a study of low-dimensional embeddings that preserve the meaningfulness of norms and small angles. These embeddings were derived from the nonlinear matrix decomposition in eq. (1), and they were learned by solving a sequence of eigenvalue problems to minimize the cost in eq. (4).

There are two clear directions for future work. The first is to investigate more scalable implementations of the algorithm in section 2. One such approach has been described in section 3, but there are many other (complementary) tools for this purpose, including the Nystrom approximation (Williams & Seeger, 2001; Drineas & Mahoney, 2005; Gittens & Mahoney, 2016), randomized algorithms (Mahoney, 2010; Halko et al., 2011; Tropp et al., 2017), and alternating least-squares methods (Hastie et al., 2015). The second direction is to develop the ideas in this paper for deep learning, as has been done for other models of manifold learning (Jia et al., 2015; Chen et al., 2018; Mishne et al., 2019; Du et al., 2021; Pai et al., 2022). In particular, we note that a simple modification of eq. (2) yields a *parametric* ReLU model for learning similarity-preserving mappings $\boldsymbol{x} \to \boldsymbol{y}$. Such a mapping is described by the implicit equation

$$\max(0, \boldsymbol{x}{\cdot}\boldsymbol{\mu}_j - \tau\|\boldsymbol{x}\|\|\boldsymbol{\mu}_j\|) \,=\, \max(0, \boldsymbol{y}{\cdot}\boldsymbol{\nu}_j - \tau\|\boldsymbol{y}\|\|\boldsymbol{\nu}_j\|), \tag{16}$$

where the weight vectors $\boldsymbol{\mu}_j$ and $\boldsymbol{\nu}_j$ are prototypes that tile, respectively, the spaces of high-dimensional inputs and low-dimensional outputs (Sengupta et al., 2018). It seems fruitful to investigate eq. (16) as a layerwise prescription for unsupervised learning in deep networks (Hinton & Salakhutdinov, 2006; Hinton et al., 2006; Mazumdar & Rawat, 2019b).

### Acknowledgements

The author thanks the anonymous reviewers for their detailed and constructive feedback. He also acknowledges many illuminating discussions with S. Seung, K. Luther, R. Yang, A. Sengupta, D. Chklovskii, and Y. Ma.

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

# A   Supplementary details for alternating minimization

In this appendix we provide further details of the alternating minimization for TSM in section 2.2, specifically its initialization, regularization, and convergence.

## A.1   Initialization

As discussed in section 2.2, the results from the alternating minimization of eq. (4) can depend on how the procedure is initialized. For the results in this paper, we adopted a simple approach, initializing this procedure from the baseline embeddings computed by SVD. We describe this initialization next.

To perform the updates in eqs. (5–6), it is necessary to initialize either the matrix $\mathbf{L}$ or the matrix $\mathbf{Z}$. We initialized the matrix $\mathbf{L}$ in the following way. Let $d$ denote the desired rank of the matrix $\mathbf{L}$. We began by setting

$$L_{ij} = \boldsymbol{v}_i \cdot \boldsymbol{v}_j - \tau \|\boldsymbol{v}_i\| \|\boldsymbol{v}_j\|, \tag{17}$$

where $\boldsymbol{v}_i$ are the $d$-dimensional projections of the inputs from eq. (10). Note that this initialization generally produces a matrix $\mathbf{L}$ of rank $d+1$; nevertheless, it seemed to work well in practice. The minimization of eq. (4) proceeds by updating $\mathbf{Z}$ from eq. (5), then updating $\mathbf{L}$ from eq. (6), and so on. The rank of $\mathbf{L}$ is therefore equal to $d$ after all subsequent updates.

As in all non-convex optimizations, it is problematic when results are sensitive to different initializations. In this respect, TSM is similar to other popular algorithms for nonlinear dimensionality reduction, such as t-SNE (van der Maaten & Hinton, 2008) and UMAP (McInnes et al., 2018), whose solutions also depend on how their non-convex optimizations are initialized. These approaches generally achieve their best results by employing specialized heuristics, such as early exaggeration, or by initializing their optimizations from the bottom eigenvectors of a graph Laplacian.

## A.2   Regularization

There can be other ways besides initialization to address the problem of local minima in non-convex optimizations. For TSM, we imposed an additional linear constraint[6] on the matrix $\mathbf{Z}$ to guide the optimization to better minima. This constraint was based on the observation that distances between dissimilar inputs tend to increase when they are faithfully embedded into a lower-dimensional space (Weinberger & Saul, 2006). Based on this intuition, we sought to encourage embeddings $\boldsymbol{x}_i \to \boldsymbol{y}_i$ satisfying

$$\sum_{ij} \left[ \boldsymbol{y}_i \cdot \boldsymbol{y}_j - \tau \|\boldsymbol{y}_i\| \|\boldsymbol{y}_j\| \right] \geq \sum_{ij} \left[ \boldsymbol{x}_i \cdot \boldsymbol{x}_j - \tau \|\boldsymbol{x}_i\| \|\boldsymbol{x}_j\| \right]. \tag{18}$$

Eq. (18) states that pairs of outputs are *on average* further apart in angle than their corresponding pairs of inputs. We encouraged solutions with this property by identifying the left side of eq. (18) with the elements of $\mathbf{Z}$ and constraining their overall sum. In particular, we replaced eq. (4) with the constrained optimization

$$\min_{\mathbf{L},\mathbf{Z}} \|\mathbf{L} - \mathbf{Z}\|_F^2 \text{ such that } \left\{ \begin{array}{rcl} \text{rank}(\mathbf{L}) & = & d, \\ \max(0, \mathbf{Z}) & = & \mathbf{S}, \\ \sum_{ij} Z_{ij} & \geq & \sum_{ij} [\boldsymbol{x}_i \cdot \boldsymbol{x}_j - \tau \|\boldsymbol{x}_i\| \|\boldsymbol{x}_j\|]. \end{array} \right. \tag{19}$$

The original update for $\mathbf{Z}$ requires only a slight modification to enforce this extra constraint. To proceed, we rewrite the original update in eq. (5) with a placeholder variable $\tilde{\mathbf{Z}}$. In particular, let

$$\tilde{Z}_{ij} = \left\{ \begin{array}{ll} S_{ij} & \text{if } S_{ij} > 0, \\ \min(0, L_{ij}) & \text{otherwise.} \end{array} \right. \tag{20}$$

---

[6]Additional constraints narrow the search space, and thus in general they may restrict solutions to *worse* minima. But a *well-chosen* constraint can also narrow the search space in a way that guides non-convex optimizations to better minima. This is what appeared to occur in practice.

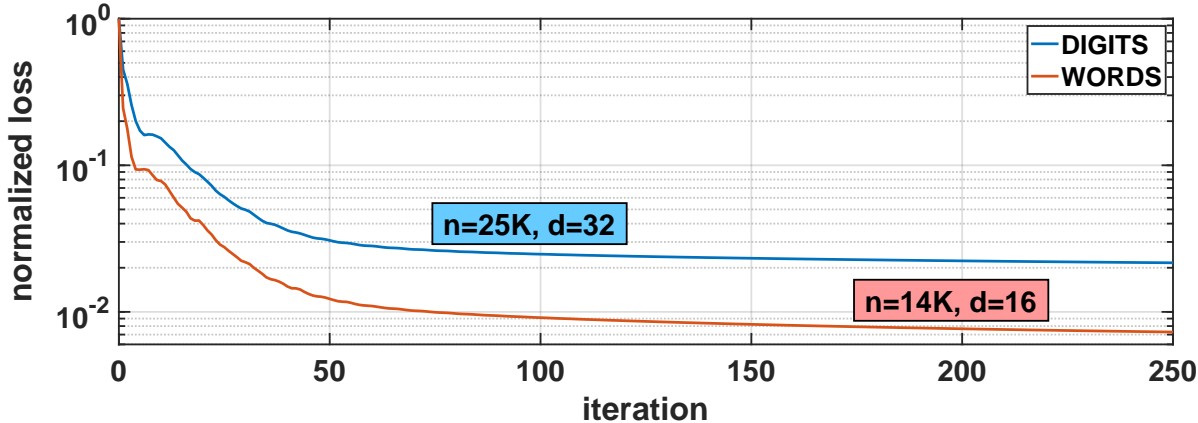

Figure 8: Behavior of the objective function in eq. (4), normalized by its initial value, over the course of 250 alternating updates to estimate **Z** and **L**. The updates were performed with the initialization, regularization, and acceleration described in eqs. (17–23).

Only one further calculation is required for the modified update. Let $n_0$ denote the number of *zero* elements in **S**, and let $\delta$ denote the amount, per zero element of **S**, by which the original update for **Ž** violates the new constraint in eq. (19):

$$\delta = \frac{1}{n_0} \max \left( 0, \sum_{ij} \left[ \boldsymbol{x}_i \cdot \boldsymbol{x}_j - \tau \|\boldsymbol{x}_i\| \|\boldsymbol{x}_j\| - \tilde{Z}_{ij} \right] \right). \tag{21}$$

Then in terms of these quantities, the update for the constrained optimization in eq. (19) takes the simple form:

$$Z_{ij} = \begin{cases} S_{ij} & \text{if } S_{ij} > 0, \\ \min(0, L_{ij}) + \delta & \text{otherwise.} \end{cases} \tag{22}$$

Comparing eq. (20) and eq. (22), we see that the constrained update differs only by a uniform shift of those elements in **Z** that are not already determined by the nonzero elements of **S**.

### A.3   Acceleration

The alternating updates in eq. (6) and eq. (22) are guaranteed to monotonically decrease the cost in eq. (19) at each step. This is an especially useful guarantee for debugging a first implementation of the algorithm. In practice, though, we found it useful to modify the update for **Z** with an additional momentum term (Polyak, 1964). In particular, let $\mathbf{Z}_t$ denote the value of **Z** after the $t^{\text{th}}$ update, and let $\mathbf{Z}_{t+1}^{\text{orig}}$ be the updated value that would have been computed originally from eq. (22). Instead, we set

$$\mathbf{Z}_{t+1} = \mathbf{Z}_{t+1}^{\text{orig}} + \gamma(\mathbf{Z}_t - \mathbf{Z}_{t-1}), \tag{23}$$

where $\gamma \in (0, 1)$ is a so-called momentum parameter. In practice, we found that the update in eq. (23) not only converges more quickly, but also finds better solutions. All of the results in section 2 were obtained by setting $\gamma = 0.9$ and completing 250 of these alternating updates. Fig. 8 plots the objective function in eq. (4), normalized by its initial value, over the course of these updates for the data sets in section 2.3.

## B   Supplementary results for Isomap

Many algorithms have been proposed for learning nonlinear neighborhood-preserving embeddings. These include several methods, such as Isomap (Tenenbaum et al., 2000), Hessian LLE (Donoho & Grimes, 2003), and maximum variance unfolding (Weinberger & Saul, 2006), that are similar in important respects to TSM.

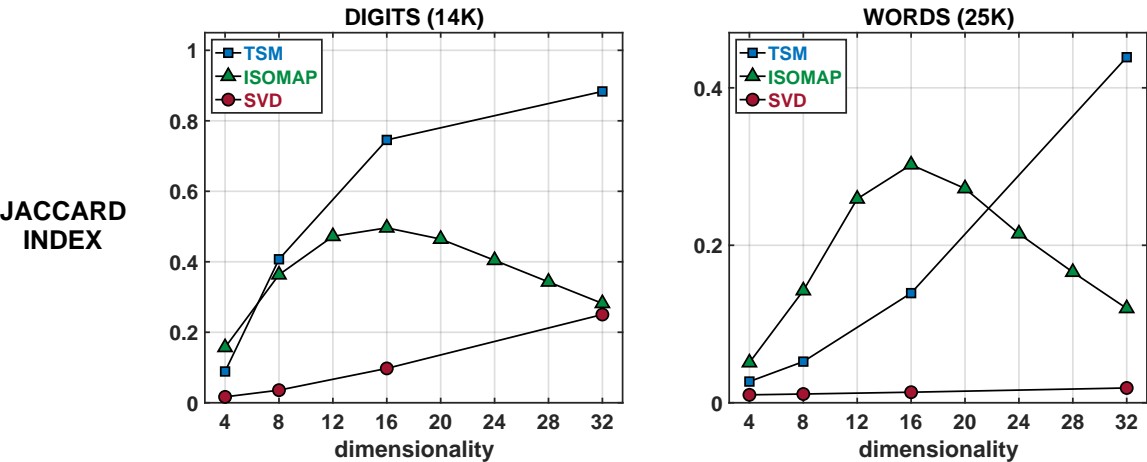

Figure 9: Jaccard indices in eq. (12) for the embeddings from **TSM**, **Isomap**, and **SVD** of 14K digit images (*left*) and 25K word vectors (*right*), as a function of their dimensionality.

In particular, the embeddings from these algorithms are designed (and in some settings, guaranteed) to preserve distances between nearby inputs; they are also derived from the eigenvectors of specially constructed $n \times n$ matrices. It is therefore instructive to compare embeddings with these properties to those of TSM.

In this appendix we compare embeddings from TSM and Isomap on the data sets of 14K images and 25K word vectors in section 2.3. We chose Isomap as a benchmark because it does not share the numerical difficulties of Hessian LLE for high dimensional data, and also because it scales better than maximum variance unfolding to data sets with large numbers of examples. It is important to emphasize that the embeddings of Isomap are *not* designed to preserve norms or angles between nearby inputs; this has important consequences, as we observe below. Nevertheless, one can still evaluate how well Isomap succeeds in computing a neighborhood-preserving embedding when it operates on the same sparse neighborhood graph as TSM.

The Isomap algorithm consists of three steps. First, it constructs a sparse graph whose nodes represent inputs and whose edges (connecting nearby inputs) are weighted by pairwise distances. Second, it uses dynamic programming to compute all shortest pairwise distances through this graph. Finally, from the $n \times n$ matrix of pairwise distances, it uses multidimensional scaling to recover a low-dimensional embedding. More specifically, this is done by assuming that the low-dimensional outputs are centered at the origin, then deriving and diagonalizing their Gram matrix.

For the experiments in this section, we seeded Isomap with a sparse graph whose edges were in one-to-one correspondence with the nonzero elements of the sparse matrix **S** computed by TSM in eq. (3). In addition, we weighted the edges of this graph by the Euclidean distances $\|\hat{\boldsymbol{x}}_i - \hat{\boldsymbol{x}}_j\|$ between nearby *normalized* inputs, where $\hat{\boldsymbol{x}}_i = \boldsymbol{x}_i / \|\boldsymbol{x}_i\|$. The Isomap embeddings were therefore designed to co-locate pairs of outputs whose corresponding inputs have small cosine distances $\frac{1}{2}\|\hat{\boldsymbol{x}}_i - \hat{\boldsymbol{x}}_j\|^2 < 1 - \tau$. Equivalently these are the pairs of inputs for which $\cos(\boldsymbol{x}_i, \boldsymbol{x}_j) > \tau$.

As before, we used a Jaccard index between sets $\Omega_x$ and $\Omega_y$ to measure the quality of the embeddings $\boldsymbol{x}_i \to \boldsymbol{y}_i$ obtained in this way; see eq. (12). For Isomap, however, we constructed the set $\Omega_y$ in a slightly different way; namely, we took $\Omega_y = \{(i,j) \,|\, \frac{1}{2}\|\boldsymbol{y}_i - \boldsymbol{y}_j\|^2 < 1 - \tau\}$. This was done to account for the fact that Isomap attempts to preserve distances rather than angles. Note that if the embedding by Isomap were perfectly neighborhood-preserving, then this set $\Omega_y$ would contain exactly those pairs $(i,j)$ that index nonzero elements of the matrix **S** in eq. 3. With this allowance, we can therefore compare the Jaccard indices from Isomap, TSM, and SVD in a meaningful way.

Figure 9 compares the Jaccard indices from these algorithms for embeddings of different dimensionality. There are two striking results. First, the embeddings from Isomap have higher (better) Jaccard indices than TSM at low dimensionalities; second, after a certain point, the higher-dimensional embeddings from Isomap

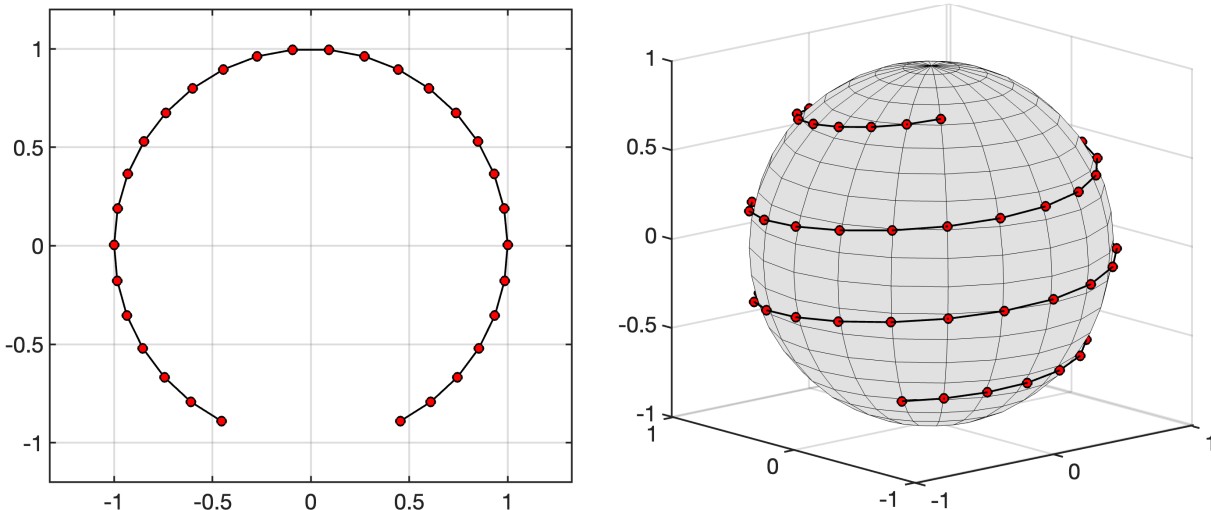

Figure 10: Two data sets that illustrate the additional requirements of norm-preserving embeddings. In these data sets, the inputs (shown in **red**) all have unit norm, and the nearest neighbors are connected by black edges. The inputs can be faithfully mapped into one dimension if the only goal is to preserve distances between nearest neighbors. But no dimensionality reduction is possible if the mapping must also preserve the norms of the inputs.

yield progressively lower (worse) Jaccard indices—a seemingly counterintuitive result. Next we show how to reconcile these observations.

First we consider why Isomap's embeddings have higher Jaccard indices for lower dimensionalities. One plausible explanation is that Isomap only aims to compute a neighborhood-preserving embedding; unlike TSM, it is not also required to compute a *norm*-preserving embedding. The latter may be difficult or impossible to accomplish when the target dimensionality is small. The additional requirements of norm-preserving embeddings are illustrated by the two panels of Fig. 10. Each panel shows a data set whose inputs, all of unit norm, are distributed like beads along a string with two open ends. In both cases, Isomap can faithfully embed the inputs into $\Re^1$ as uniformly spaced points along the real line. But while such mappings can preserve distances between nearest neighbors, they cannot also preserve distances to the origin. Such effects may account for the lower Jaccard indices of TSM at lower dimensionalities, as its embeddings in this regime are subject to more stringent constraints.

Next we examine why Isomap's embeddings of these data sets appear to deteriorate—at least, as measured by their Jaccard indices—for higher dimensionalities. Here the most plausible explanation is that the data sets violate a basic assumption of the Isomap algorithm; the algorithm's guarantees hold only when the underlying manifold is isometric to a convex subset of Euclidean space. In practice this breakdown is signaled by the presence of negative eigenvalues in the matrix that Isomap diagonalizes. Figure 11 plots these eigenvalues for the data sets of 14K digits and 25K word vectors. For each data set, the first negative eigenvalue appears at a dimensionality that slightly precedes the onset of decreasing Jaccard indices in Fig. 4. The negative eigenvalues also increase in frequency as the positive eigenvalues decrease in magnitude. This suggests a plausible explanation for the decreasing Jaccard indices of Isomap at higher dimensionalities.

It should be noted that we have applied Isomap in an unconventional way—normalizing the inputs to unit length before computing distances—in order to permit a direct comparison to TSM. Recognizing this, we believe that the results in this section best serve to highlight the different requirements of norm-preserving embeddings, as well as the different approaches needed to compute them.

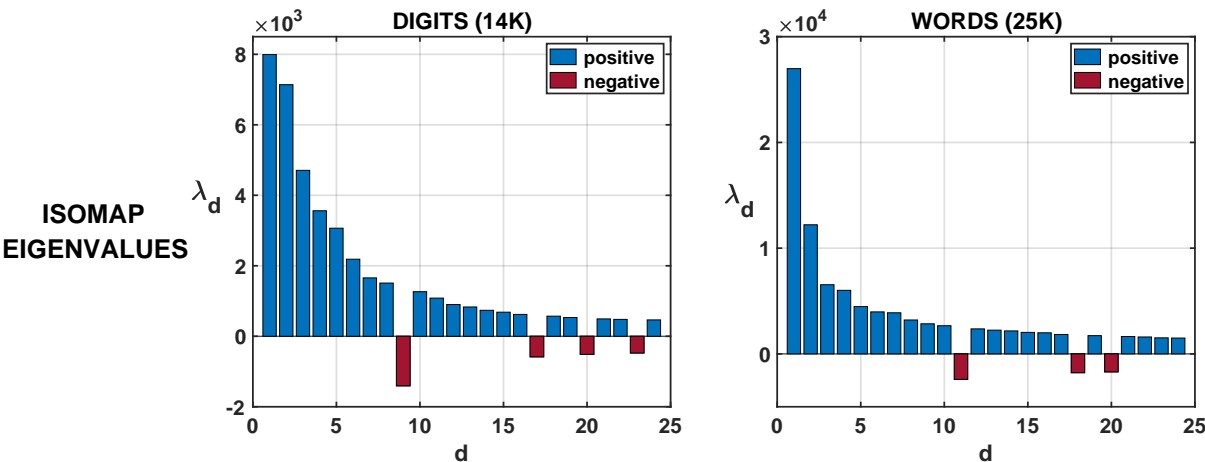

Figure 11: Eigenvalues of the Gram matrix diagonalized by Isomap. Negative eigenvalues (shown in **red**) indicate that Isomap's graph distances do not correspond to distances in any Euclidean space.

## C Supplementary details for LLX

Here we give further details to solve the sparse least-squares problem in section 3.1. As shorthand, let $\mathbf{Y} = [\boldsymbol{y}_1\,\boldsymbol{y}_2\,\ldots\,\boldsymbol{y}_n]$ denote the outputs of the embedding, and let $\mathbf{I}$ denote the $n \times n$ identity matrix. In this notation, we can rewrite eq. (15) as

$$\mathcal{E}_{\mathbf{W}}(\mathbf{Y}) = \mathrm{Trace}\Big[\mathbf{Y}(\mathbf{I}-\mathbf{W})^{\top}(\mathbf{I}-\mathbf{W})\mathbf{Y}^{\top}\Big]. \tag{24}$$

To proceed, we distinguish the $m$ known outputs of $\mathbf{Y}$ (i.e., the landmarks) previously computed by TSM versus the $n-m$ unknown outputs that remained to be computed by LLX. Without loss of generality, and to simplify notation, we assume that the data is ordered such that the $m$ landmarks correspond to the *last* examples of the data set. Then we can write

$$\mathbf{Y} \;=\; \begin{bmatrix} \mathbf{Y}_{\mathrm{u}}\ \mathbf{Y}_{\mathrm{k}} \end{bmatrix} \quad \text{where} \quad \begin{cases} \mathbf{Y}_{\mathrm{u}} = \begin{bmatrix} \boldsymbol{y}_1\ \boldsymbol{y}_2\ \cdots\ \boldsymbol{y}_{n-m} \end{bmatrix} & (\text{unknown}), \\ \mathbf{Y}_{\mathrm{k}} = \begin{bmatrix} \boldsymbol{y}_{n-m+1}\ \boldsymbol{y}_{n-m+2}\ \cdots\ \boldsymbol{y}_n \end{bmatrix} & (\text{known}). \end{cases} \tag{25}$$

We continue in this way by decomposing the $n \times n$ matrix $\mathbf{I}-\mathbf{W}$ inside the quadratic form of eq. (24). Specifically, we write this matrix as

$$\mathbf{I} - \mathbf{W} \;=\; \begin{bmatrix} \mathbf{A} & \mathbf{B} \\ \mathbf{C} & \mathbf{D} \end{bmatrix} \quad \text{where} \quad \begin{array}{l} \mathbf{A} \in \Re^{(n-m)\times(n-m)}, \\ \mathbf{B} \in \Re^{(n-m)\times m}, \\ \mathbf{C} \in \Re^{m\times(n-m)}, \\ \mathbf{D} \in \Re^{m\times m}. \end{array} \tag{26}$$

Note that on the right side of this equation, all of the submatrices $\mathbf{A}, \mathbf{B}, \mathbf{C}, \mathbf{D}$ are sparse—a property they inherit from the matrices $\mathbf{I}$ and $\mathbf{W}$ on the left. With this notation, we can explicitly write out the linear equations that locate the minimum of eq. (24). In particular, we have:

$$\Big[\mathbf{A}^{\top}\mathbf{A} + \mathbf{C}^{\top}\mathbf{C}\Big]\mathbf{Y}_{\mathrm{u}}^{\top} \;=\; -\Big[\mathbf{A}^{\top}\mathbf{B} + \mathbf{C}^{\top}\mathbf{D}\Big]\mathbf{Y}_{\mathrm{k}}^{\top} \tag{27}$$

In this way we obtain $d$ systems of sparse linear equations for the $d$ rows of $\mathbf{Y}_{\mathrm{u}}$. In practice, we do not solve these equations by inverting the $(n-m) \times (n-m)$ matrix on the left side of eq. (27). Instead, we solve each system independently using a preconditioned conjugate gradients method (Barrett et al., 1994). This latter approach is much faster.

