# OpenReview forum: "A geometrical connection between sparse and low-rank matrices and its application to manifold learning"
_TMLR — Accepted by TMLR_

### Review · Reviewer_KAJF · 2022-10-02

**Summary Of Contributions:**

For a set of $N$ high-dimensional points given as the columns of matrix $X\in \mathbb{R}^{d\times N}$, define $\bar{G}(X) \coloneqq [ \langle x_i , x_j \rangle - \kappa \|x_i\| \|x_j\| ]\_{ij}$ and consider a *thresholded similarity matrix* given by $S_\kappa(X) \coloneqq \max(0, \bar{G}(X) ) $.

The authors propose to find low-rank matrices $L$ and $Y$ such that first, $S_\kappa(X)  \simeq_{\rm ls} \max(0, L)$, and second, $L \simeq_{\rm ls} \bar{G}(Y)$, as a means for *embedding* $X$ into a lower-dimensional space (as $Y$).  Note that:

Considering $S \simeq \max(0, L)$, observe that ``the sparser the matrix S, the larger the space that can be explored to discover a matrix L of significantly lower rank.`` Therefore, for a given sparse matrix $S$, one could be hopeful that a low-dimensional matrix $L$ in the above exists, satisfying $S \simeq \max(0, L)$. The reasoning behind the ``second`` part in the above being possible is less complete. Note $L$ and $Y$ are not being estimated simultaneously, rather first $L$ then $Y$ from $L$.

The authors discuss the interrelated roles of $\kappa$ and the rank chosen for the final solution when estimating $L$, but do not provide algorithmic proposals for their simultaneous selection; especially in relation to possible properties of data ($X$). Various useful discussions on the role of $\kappa$ has been given.

The authors propose an alternating minimization procedure (non-convex, based on least-squares) for finding $L$ and a least-squares procedure for estimating $Y$ from $L$. The former does not have any formal guarantees on convergence or on the rate of convergence.

The authors provide useful numerical experiments, as well as an ad-hoc procedure (LLX) for applying the proposed method to large data sets (as the proposed method suffers from memory issues due to SVD for $N\times N$ matrices).

**Broader Impact Concerns:**

Probably a discussion considering the analogy completion part.

**Requested Changes:**


---
---

Remark 0: The descriptions and discussions in Section 2.1 (Model) are excellent; clear and complete. However, I have various comments about the summaries and interpretations of this model given before Section 2.1, as mentioned below; especially the content of page 2 starting from ``But eq. (1) by itself has several limitations, ...`` till the end of the page.


---


The problem description in the abstract (``We consider when a sparse nonnegative matrix $S$ can be recovered from a real-valued matrix $L$ of significantly lower rank``) is rather vague and significantly more general than $S=\max(0,L)$ in Eq 1, and more specifically than the setup studied in the paper in Eq 2; especially note ``can be recovered from``. In fact, if Eq 2 is the limit of the scope of the current study, I think it should be mentioned from the very beginning as the problem description.

---

Instead of ``Many problems in high dimensional data analysis can be formulated as a search for structure in large matrices``, and considering the two examples given right after it, I think it is more appropriate to say ``Many problems in high dimensional data analysis can be formulated as a search for large structured matrices.``

---

Bottom of page 1: Considering ``The intuition behind this observation is illustrated in the left panels of Figure 1.`` I think the left and middle panels of Figure 1 provide an illustration for Eq 1 and not for the implication claimed in ``the sparser the matrix $S$, the larger the space ...``. For the latter, it would be informative to use a sparsified version of $S$ (in Fig 1) and provide a matrix $L$ of *least rank* (and the same for $S$ itself, to compare); where the *least rank* property has been insured by exhaustive search (going to a smaller dimension for $S$ might help with computations; both to find such a least rank matrix, and to possibly have a unique least rank matrix so that the comparison is possibly more informative.)

---

In the middle of page 2, please elaborate on Item 2; if `sparsification' (by thresholding) of the true similarity matrix is meant (as discussed in the third paragraph of page 10), I think it should be stated as a step enabled by the properties of the data (e.g., see Remark 2 and Remark 3) rather than a general way of dealing with non-sparse data (of course identifying and exploiting such property of the data is a contribution, and I am pointing out that it would be helpful if it is clarified).

In addition, I am not sure if I understand what the authors mean by ``a geometrical picture`` for ``a dataset``, in Item 1. After the three items, could you clarify what the ``inputs`` mean in ``The inputs are not required to be sparse``?

---


As discussed on top of page 4, in Section 2.1, the paper considers manifold learning when the origin and angles are meaningful. I suggest that this distinction with manifold learning in general be mentioned in lines 3-5 of the abstract (I understand that it is somewhat 'apparent' in the rest of the abstract), and especially in the paragraph on top of page 3 (the paragraph before last, in Section 1.)


---

In the last paragraph of Section 1, in ``input similarities are encoded by a sparse matrix $S$ of thresholded inner products.`` I am not sure if ``input similarities are encoded by`` is exactly a correct description; maybe ``input data is in the form of thresholded input similarities`` instead?



---


In the caption of Figure 2, adding *1NN' to ``nearest neighbors`` (other than the legend) might be helpful to the readers.



---
---

Let me provide a definition, aligned with the goal of paper in studying manifold learning in setups where origin has a meaning (top of page 4 in Section 2.1). For any nonzero point $x\in\mathbb{R}^d$, define $\bar{x}\coloneqq x/\|x\| \in S^{d-1}$ where $S^{d-1}$ is the unit sphere in $\mathbb{R}^d$.

**Definition 1:** Given a **similarity-encoding** function $P: \mathbb{R}^{d}\times \mathbb{R}^{d} \to \mathbb{R}_+$, for each matrix $X\in \mathbb{R}^{d\times N}$ with no zero columns, define a **$P$-refined similarity matrix** as
$$
S_P (X)  \coloneqq  [ \| x_i \| \cdot \| x_j \| \cdot P( x_i , x_j  )  ]_\{ij} \in \mathbb{R}^{N \times N}
$$
Here $P$ indicates **a refined similarity** provided by **a pre-processing step**; e.g., `Step 1' of Section 2.2, or a 1-nearest-neighbor procedure discussed at the end of Section 1.

**Example 1:** For a given $\kappa\in (0,1)$, consider $P^\kappa$ defined as
$$
P^\kappa ( x_i , x_j ) \coloneqq \max ( 0 , \cos( x_i , x_j )  - \kappa )  = ( \cos( x_i , x_j )-\kappa ) \cdot \delta\_{ij} ,
$$
where $\delta_{ij}$ is a 0/1 indicator function for whether $\cos(x_i,x_j)$ exceeds $\kappa$.
Note that $P^\kappa$ is the output of the pre-processing step in `Step 1' of Section 2.2.

**Example 2:** For a given $\kappa\in (0,1)$, consider $P^{\kappa, \rm nn}$ defined as
$$
P^{\kappa, \rm nn}( x_i , x_j ) \coloneqq (\cos(x_i, x_j)-\kappa) \cdot p\_{ij} \,,
$$
where $p_{ij}$ is a 0/1 indicator function for whether $x_j$ is the nearest neighbor to $x_i$ or not. Note that we are slightly abusing the notation, by allowing $P$ to have access to all of $X$.

**Example 3:** For a given $\kappa\in (0,1)$, consider $P^{\kappa, \rm deb}$ defined as
$$
P^{\kappa,\rm deb} ( x_i , x_j ) \coloneqq \cos(x\_i, x\_j) \cdot \delta\_{ij} \,,
$$
where $\delta_{ij}$ is a 0/1 indicator function for whether $\cos(x_i,x_j)$ exceeds $\kappa$. Note that $P^{\kappa,\rm deb}$ is a ``debiased`` version of $P^\kappa$ in Example 1.


---

Remark 1: There are two separate notions that are being used interchangeably in parts of the text.
One is the **thresholded similarity matrix** given by $S_{P^\kappa}$,
and the other is the **local similarity matrix** given by $S_{P^{\kappa, \rm nn}}$.
I think it would be helpful to distinguish between these two and refer to the appropriate one when needed. I will refer to this remark throughout my review.

---

``The algorithm regards the inputs $x_i$ and $x_j$ as similar whenever the cosine of the angle between them exceeds some threshold`` -- any relations (setup or method) to TSC (Thresholded Subspace Clustering; arXiv:1303.3716)?

---

To bring up a possibility: Thresholded (an operation, namely $S_{P^\kappa}$ as defined in Remark 1, and as performed in Eq 2) Similarity Matching in stead of Sparse (a property, which is not present for neither $x_i$'s or $y_i$'s, unless an operator acts on them) Similarity Matching.

---

Note: in this notation, the use of $S_{P^\kappa}$ and the ``de-biased`` version $S_{P^{\kappa, \rm deb}}$ seem to be equivalent.

---
---



Remark 2:
Consider Remark 1. With $\kappa$ chosen based on a `\%99' threshold, for the two specific datasets mentioned here, $S_{P^\kappa}$ is approximately the **weighted 1-NN graph** of the original data (i.e., $S_{P^\kappa}(X) \simeq S_{P^{\kappa, \rm nn}}(X)$), and the proposed method, namely

> ``(Task 1:) seeking a low-dimensional embedding of the given high-dimensional data whose `thresholded similarity matrix' matches that of the original data``,

is close to

> ``(Task 2:) seeking a low-dimensional embedding of the given high-dimensional data whose `weighted 1-NN graph' matches that of the original data``.

Could you please discuss

- (Q1) whether other approaches for `Task 2` exist in the literature,

- (Q2) what are the corresponding *geometric insights* behind those algorithms that might differ from the geometric picture (Eq 1, etc) behind your model (the three items in Section 2.1, or equivalently in Eq 2)?
I am asking this because while the model of Section 2.1 for the two datasets and the choice of `\%99` matches `Task 2` to a good extent, it would deviate from it in other scenarios (other data, other $\kappa$), hence is expected to correspond to a different underlying *geometric picture* compared to other methods addressing `Task 2`.

- (Q3) are there data and $\kappa$ settings, of practical interest, where `Task 1` and `Task 2` diverge?

---
---

Remark 3:
The following are some questions on the proposed method, at the end of Section 2.1, for choosing $\kappa$.

- (Q1) Given $N$ points, there are $N$ angles between each point and its nearest neighbor and $N(N-2)$ angles between each point and its non-nearest neighbors. As a side point, I suggest clarifying that the blue and the orange histograms in Figure 2 have been normalized `separately'. Do the authors have any arguments as to why these two separately-normalized histograms, whose underlying data have sizes of different orders of magnitude, are comparable in the sense that their dividing point can be meaningfully defined, depicted, or used?

- (Q2) Do the authors believe that a threshold such as `\%99` would generally be close to the dividing point of the two histograms? (say for data sets that have been already deemed `suitable' for the proposed method by confirming that their two histograms are mostly non-overlapping; such as the two datasets here)


- (Q3) When a dataset has two nearly-non-overlapping histograms, can we say that the dataset is **easy-to-embed** in some sense and proceed with a different method than the proposed one which forms $S_{P^\kappa}(X)$? For example, *does the information that the farthest nearest neighbor is in less distance than the closest second-nearest neighbor allow for a simpler embedding procedure?* (basically, I am curious about any contenders from the literature in such a `simple' setup.)

- (Q4) Could you reconcile the other points in this remark with the proposal at the end of page 4; ``Note that if this situation occurs for more than a small fraction of inputs, then it suggests to use a smaller value of $\kappa$ in eqs. (2–3).`` In other words, is there a conflict between a dataset satisfying the above and it requiring lots of `virtual points`, and if not, then for $\kappa$ chosen using the `\%99`-procedure for datasets such as those above, what would reducing $\kappa$ mean in relation to the histograms.

- (Q5) Could you reconcile the other points in this remark with the interrelation of $\kappa$ and $d$, discussed at the end of Section 2.2, especially in relation to the properties of the Gram matrix. (beyond the empirical approach in ``This question can be answered to some degree by empirical evaluations, to which we turn next`` and Section 2.3, and instead, possibly using the classification of regimes offered by the *non-overlapping* property or the `Task 1` $\simeq$ `Task 2` property from the above).

---
---

Section 2.2, bottom of page 5: please provide a formal discussion on why the method converges; i.e., why the *reduction* is strict, etc.

---

Section 2.2, bottom of page 5: it would be helpful to clearly state that the authors do not have a guarantee on the rate of convergence and the proposed algorithm could take a long time to converge in some scenarios, in addition to pointing to convergence plots for the datasets and values of $\kappa$ on which they have already experimented.


---

Section 2.2, middle of page 6, on ``The above solution was based on an assumption ... When this is not the case``: I believe the non-existence happens if and only if $G$ in Eq 8 is not positive semidefinite, and, the result is not desirable when $G$ in Eq 8 has a high rank. Could you please discuss these situations further (linear algebraic and Eq 8; see my *Summary Of Contributions* and what is mentioned as ``second``), before moving on to a least-squares approach?

---

I think the nonlinearity present in the specification of the model makes SVD irrelevant as a baseline (as clear from Table 1). Possibly, existing algorithms for `Task 2` (see Remark 3, Q3) could be used as baselines.

---

In Section 3, how do we select the landmarks?


**Strengths And Weaknesses:**

I believe the paper is very clear, to the point, and (to a good extent) complete in its discussions, for the setup under study. However, I would still like to ask the authors to address the points I bring up, before publication, especially I would like to see clarifications on the *modeling assumptions* here; situations in which they are satisfied and places they show up in the paper (in the arguments, interpretations, etc).

---

### Review · Reviewer_ejvv · 2022-10-04

**Summary Of Contributions:**

The paper presents a new optimization problem for learning low rank representations of data. In particular, the paper is interested in learning low rank representations of sparse data. The paper then presents a method for learning representations by attempting to solve the optimization problem. Finally, they demonstrate the usefulness of the learned representation on two datasets - MNIST and Word Embeddings. The paper demonstrates superiority to the SVD on two different statistics.

**Broader Impact Concerns:**

No broader impact concerns.

**Requested Changes:**

I have two major change requests.

1) I think convergence issues of the algorithm (weaknesses 1,2) should be talked about more. Hence expanding on this would be good.

2) Providing more background on the equation $S \approx \max(0,L)$ would also be appreciated.

**Strengths And Weaknesses:**

**Strengths**
---

1) I think the idea of building connections between sparse matrices and low rank matrices is very interesting. Both concepts are very important in learning low complexity representations. So developing methods that can convert one representation to the other is very interesting.

2) The experiments are interesting in the following sense. To me whenever, we learn lower dimensional representations of data, the holy grail of statistics to preserve is some metric (need not be the Euclidean metric). In this case, however, the paper suggests, ignoring the Euclidean metric, but instead focusing on the cosine similarity measure and well as preserving the norms of the data points.

**Weakness**
---

For the me the major weakness revolves around theoretical considerations of the problem

1) The problem is non-convex. While this is clear due to the constraints. This is not discussed until the appendix. I think a discussion about this would be needed in the paper.

2) Convergence of the method. I think some more work needs to be done to talk about the convergence of the method. Specifically, at the bottom of page 5, it says "By construction, each of the updates in eqs. (5–6) is guaranteed to reduce the value of the objective function in eq. (4). Hence they can be alternated until they reach a desired level of convergence." Either a proposition/theorem with this result or a reference would be good.

In particular, this method has the same flavor as many different projection based optimization techniques such a Dykstra method of alternate projections. While the space of rank at most d matrices is not convex, the projections (modulo having repeated singular values) is well defined. So some theory may apply to provide convergence (not necessarily to the optimal).

Also why look at $\|L - Z\|_F^2$ as the objective and not $\|S - \max(0,L)\|_F^2$ as the objective. I feel like these might be equivalent.

3) I think more discussion on Equation (1) would be appreciated. In particular, what does the approximation mean here and are there results formally connecting sparsity of $S$ to the rank of $L$.


**Questions**
---

1) Do you know anything about the true optimal solution for the problem? Maybe some approach that looks at Lagrange multiplier might yield some results?

2) While the metric is not objective for the reduction. How well does it perform with respect to metric preservation? Note SVD, might not actually do that well on metric preservation (even MDS might not, if we start with non-euclidean data, but here is all about euclidean data.)

---

### Review · Reviewer_WDzC · 2022-10-04

**Summary Of Contributions:**

The authors apply an algorithm of Saul (2022) for computing a representation of
a sparse matrix as an elementwise thresholding (or a ReLU-ing, in deep learning
parlance) of a matrix of (perhaps significantly) lower rank to the problem of
nonlinear dimensionality reduction of a dataset in a high-dimensional space
$\mathbb{R}^D$. In this context, the primary contribution is to identify that
Saul's algorithm can be applied in this setting (because the data matrix itself
need not be sparse), and to evaluate the suitability of the algorithm in this
setting. The authors consider data for which the norm is meaningful (e.g., word
embeddings or images/image patches with varying illumination), and seek an
embedding into a lower-dimensional space $\mathbb{R}^d$ that preserves these norms,
as well as local information (cosine similarities) and some global information
(things that are far apart stay far apart). The embedding process has three
stages. First, the data are used to construct a matrix $S_{ij} = \mathrm{max}
\\{ \langle \boldsymbol{x}_i, \boldsymbol{x}_j \rangle - \kappa
\\| \boldsymbol{x}_i \\|\\| \boldsymbol{x}_j \\|, 0 \\}$ of thresholded similarities;
this is similar to many procedures in manifold learning, but happens to be
homogeneous in the scale of the data. Next, the algorithm of Saul is used to
obtain a rank-$d$ matrix $\boldsymbol{L}$ which satisfies $\boldsymbol{S}
\approx \mathrm{max} \\{ \boldsymbol{L}, \boldsymbol{0} \\}$, via an alternating
minimization procedure (nonconvex objective) with special initialization.
Finally, the embedded data matrix $\boldsymbol{Y}$ of $d$-dimensional vectors
is computed from the matrix $\boldsymbol{L}$ by a standard decomposition of an
associated Gram matrix. (From this description, one may infer that the
contribution of the authors' approach is to combine existing techniques into a
new approach to this problem.)
The hyperparameters $\kappa$ and $d$ above should be tuned on the data.
The authors do not provide convergence guarantees for their nonconvex approach,
but rather demonstrate experimentally that it produces sensible embeddings,
measured through mean absolute deviation of the angles of nearby points
(quantifying the degree of preservation of the local input geometry) and
the Jaccard index of the supports of $\mathrm{max}\\{\boldsymbol{S},
\boldsymbol{0}\\}$ and $\mathrm{max}\\{\boldsymbol{L}, \boldsymbol{0}\\}$
(quantifying the quality of the solution to the nonconvex optimization
procedure of Saul described above), on an image dataset (MNIST) and a language
dataset (pretrained word embeddings) versus an SVD-based baseline embedding.
The proposed method obtains embeddings whose degree of "faithfulness" to the
input geometry, measured by these metrics, grows more quickly as the embedding
dimension increases than those of the SVD baseline. The authors discuss an
incremental version of their algorithm in Section 3 that has the potential to
scale to larger datasets (e.g., $n = 10^5$) with a mild loss of representation
quality.



**Requested Changes:**

- Comparisons to other manifold learning methods (either small extensions to
  other methods to allow them to operate in the present "scale-homogeneous"
  setting, or the use of the proposed method in the setting where all data have
  the same scale) seem essential to judge whether the proposed method is
  suitable for use in other manifold learning applications, how the
  representations it learns are different from those of other manifold learning
  procedures, etc. The SVD baseline does not seem sufficient here, as it has no
  ability to capture nonlinear low-dimensional structure in the input data.
  This seems to be an essential addition for future work to build on this
  submission.

## Minor changes/clarifications

- At the bottom of page 7, this error term is between matrices that do not have
  the same size. Should it be $\boldsymbol{U} \boldsymbol{P}$ instead?
- I was slightly confused reading how the Jaccard index is used for evaluation
  on page 8. The authors refer to eqn (7) as the definition of the matrix
  $\boldsymbol{L}$, but this equation defines this matrix in terms of the
  embeddings $\boldsymbol{Y}$ that are constructed in the sequel rather than as
  the output of "Step 2", and it is not clear to me that the "if and only if"
  statements made at the end of the Jaccard index explanation hold when the
  Gram factorization procedure needs to be done approximately, as described in
  "Step 3" (in particular, when this occurs, eqn (7) only holds approximately,
  if I follow correctly). Could the authors clarify what is being measured
  here?



**Strengths And Weaknesses:**


## Strengths

- The paper is very well written -- the method is described clearly and
  completely, the work is very well situated with respect to prior work in the
  various areas the work touches, and the experimental results are clearly
  presented.
- The approach described here has some interesting potential connections to
  ReLU networks in deep learning, although the connection is left to future
  work to be explored in detail.
- The authors provide detailed guidance of various ways to tune the
  hyperparameters (Figure 2, and around Step 3/Practical Considerations
  sections on page 6), and initialize and speed up the algorithm (in Appendix
  A).

## Weaknesses

- The method does not seem to be as robust as some other manifold learning
  approaches. Although all these approaches have some amount of hyperparameter
  tuning involved (e.g., the number of nearest neighbors or neighborhood size
  to use in constructing adjacency graphs), the authors' method also has
  some brittle-feeling aspects like the use of "virtual inputs" that
  interpolate between inputs in the dataset to avoid having zero columns in
  $\boldsymbol{S}$ (bottom of page 4), the procedure used to factor the matrix
  $\boldsymbol{L}$ when the Gram matrix representation is invalid (Step 3 on
  page 6), and the seeming importance of the initialization used for good
  performance with the alternating minimization procedure (Appendix A).
  The experimental evaluation of performance under these aspects of the method
  (e.g., ablations) is mostly omitted.
- In general, the experimental evaluation is limited. Since the method is
  eventually a method for manifold learning, it would be very useful to compare
  to other manifold learning methods (not just SVD, as is done here). This
  makes it infeasible to evaluate the paper's contribution beyond simply
  introducing a new method and showing that the associated nonconvex procedure
  can be tuned to produce valid embeddings.

---

### Decision · Action_Editors · 2022-11-25

**Recommendation:** Accept with minor revision

**Comment:**

This paper considers the problem of learning a similarity-preserving mapping of high-dimensional data into a low-dimensional space. The problem is formulated as recovering a low-rank matrix whose positive elements match a sparse matrix. The authors propose an alternating minimization algorithm to solve the problem and present experiments on MNIST and pre-trained word vectors to demonstrate the performance for manifold embedding.

The findings could be of interest to researchers in manifold learning, low-dimensional models, and beyond. All the reviewers acknowledged the contribution of the paper and the submitted recommendations are (at least leaning toward) acceptance. I agree with the reviewers to recommend accepting.

There are some comments (e.g., from Reviewer WDzC) that could help improve the manuscript and the authors agree to incorporate them into the final version. The authors are advised to address them in the final version.

**Audience:**

This paper presents new methods for manifold learning and also provides a connection between two fundamental low-dimensional models, the sparse and low-rank models. The findings could be of interest to researchers in manifold learning, low-dimensional models, and beyond.

**Claims And Evidence:**

This paper considers the problem of learning a similarity-preserving mapping of high-dimensional data into a low-dimensional space. The problem is formulated as recovering a low-rank matrix whose positive elements match a sparse matrix. The authors propose an alternating minimization algorithm to solve the problem and present experiments on MNIST and pre-trained word vectors to demonstrate the performance for manifold embedding and support the claims.